# Timeliness-Aware On-Site Planning Method for Tour Navigation

**Shogo Isoda** [1,2,*], **Masato Hidaka** [1], **Yuki Matsuda** [1,2], **Hirohiko Suwa** [1,2] **and Keiichi Yasumoto** [1,2]

1   Nara Institute of Science and Technology, Nara 630-0192, Japan; hidaka.masato.hi3@is.naist.jp (M.H.); yukimat@is.naist.jp (Y.M.); h-suwa@is.naist.jp (H.S.); yasumoto@is.naist.jp (K.Y.)
2   RIKEN, Center for Advanced Intelligence Project AIP, Tokyo 103-0027, Japan
*   Correspondence: isoda.shogo.in9@is.naist.jp; Tel.: +81-743-72-5392

**Abstract:** In recent years, there has been a growing interest in travel applications that provide on-site personalized tourist spot recommendations. While generally helpful, most available options offer choices based solely on static information on places of interest without consideration of such dynamic factors as weather, time of day, and congestion, and with a focus on helping the tourist decide what single spot to visit next. Such limitations may prevent visitors from optimizing the use of their limited resources (i.e., time and money). Some existing studies allow users to calculate a semi-optimal tour visiting multiple spots in advance, but their on-site use is difficult due to the large computation time, no consideration of dynamic factors, etc. To deal with this situation, we formulate a tour score approach with three components: static tourist information on the next spot to visit, dynamic tourist information on the next spot to visit, and an aggregate measure of satisfaction associated with visiting the next spot and the set of subsequent spots to be visited. Determining the tour route that produces the best overall tour score is an NP-hard problem for which we propose three algorithms variations based on the greedy method. To validate the usefulness of the proposed approach, we applied the three algorithms to 20 points of interest in Higashiyama, Kyoto, Japan, and confirmed that the output solution was superior to the model route for Kyoto, with computation times of the three algorithms of $1.9 \pm 0.1$, $2.0 \pm 0.1$, and $27.0 \pm 1.8$ s.

**Keywords:** on-site planning; tourism recommendation; context awareness; decision making

## 1. Introduction

In recent years, demand in the tourism industry has continued to increase, as has the cost of trips taken by tourists [1]. Accompanying these increases has been an expansion of research related to personalized tourist spot recommendations (sightseeing navigation) [2–9]. Recognizing that tourist plans are often disrupted by unexpected events such as sudden heavy rain, congestion, special events, and temporary closures, leading to visitor disappointment and dissatisfaction, we propose a tourism planning approach that takes into account such unexpected events, as well as a number of additional dynamic factors, seeking to optimize the visitor's overall tourist experience. In developing this approach, we define a tourist context as the totality of the tourist situation, including the environment of the various available tourist destinations. In general, there are two types of tourist contexts: (i) static tourist contexts, which remain fundamentally unchanged such as the location, operating hours, prices, and characteristics of the tourist spots, and (ii) dynamic tourist contexts, which include weather, congestion, special events, and temporary business closures [10].

Many of the existing tourist planning systems make recommendations based on a static tourist context, which means the system's recommendations remain unchanged whether a visitor conducts

his/her search before or during a tour. In reality, however, the situation at almost all tourist spots is dynamic. For example, if a spot cannot be visited on a particular day due to a temporary holiday closure, a potential visitor's satisfaction with the spot is negated. Moreover, a visitor's level of satisfaction with a tourist spot can vary greatly depending on the weather, the level of congestion, and the presence or absence of special events such as a winter light-up. To accommodate such variables, designing a system that can collect or predict the dynamic tourist context for each tourist spot in real time and make recommendations accordingly seems highly desirable.

In addition to having the above limitations, many existing systems make recommendations based only on the satisfaction of the next spot to be visited. However, if a high-satisfaction tourist spot is located at a considerable distance, the extended travel time may limit the tourist's options after visiting that spot (e.g., fewer spots can be visited), thus reducing his/her overall satisfaction with the tour. In some cases, visiting three or more lower-satisfaction spots may lead to higher overall satisfaction than visiting a single high-satisfaction spot. It follows, then, that recommendations should take into account not just the satisfaction level associated with the next spot to visit, but also the satisfaction levels of possible visits that might follow.

A visitor's level of satisfaction with a particular spot may also differ depending on the time of day (e.g., a spot with a beautiful night view), making it important to recommend the most satisfying time of day for the spot in order to improve the overall satisfaction of tourists [11].

Some existing systems such as P-Tour [12] allow users to calculate a semi-optimal tour visiting multiple spots before starting sightseeing, but their on-site use is difficult due to the large computation time, no consideration of dynamic factors, etc.

In this paper, we propose an on-site tourism planning algorithm that considers the dynamic tourist context and the tourist's expected overall satisfaction. Identifying the tour route with the maximum expected satisfaction would require the calculation of satisfaction levels for all possible tour patterns potentially involving a huge amount of computational time. In addition, the behavior of tourists may change locally. We plan to develop a smartphone-based on-site tourism planning support application. The reason for developing a mobile application is that, even if tourists encounter unexpected events, they can take action on the spot. Our proposed approach quickly finds a quasi-optimal solution. In this study, we develop the algorithms to calculate the satisfaction level of the next spot and the expected satisfaction level of the next spot, instead of searching for the route with the highest overall satisfaction in a short time.

Specifically, we propose three algorithms, one of them is the baseline and the others are the algorithms that considered the level of expected satisfaction:

- Algorithm A (Time Series Greedy Algorithm) is a greedy algorithm that identifies, in order, the top three spots with the maximum scores, considering only the next spot.
- Algorithm B (Whole Single Greedy Algorithm) is a greedy algorithm that selects, in order, the top three spots from the pairs of spots and times with the highest scores, taking into account the overall tour time.
- Algorithm C (Whole Greedy Algorithm with Search Width) is an extended version of Algorithm B. It is a greedy algorithm that searches the choices $k$ by $k$ in a tree structure.

In order to evaluate the usefulness of the three proposed algorithms, we applied each of them to 20 points of interest (PoIs) in Higashiyama, Kyoto, Japan. The main contributions are summarized as follows:

- First, we design three algorithms for on-site tourism planning that takes into account the dynamic tourist context and the expected satisfaction of tourists in the future (Section 4).
- Secondly, we apply the three proposed algorithms to 20 PoIs in Higashiyama, Kyoto, Japan (season: autumn) to confirm the effectiveness and on-site of the proposed algorithms. Compared with the tourist routes published in Kyoto tourist magazines, Algorithms B and C confirm that each spot could be visited at a time of high satisfaction (Section 6).

- Finally, we compare the method of recommending based on the satisfaction with the next spot (Algorithm A) and the method of recommending based on the expected satisfaction of the next and subsequent spots (Algorithms B and C) in terms of total satisfaction and number of spots visited. We confirm that overall satisfaction and the number of spots to be visited will be improved by taking into account the level of expected satisfaction (Sections 6 and 7).

The rest of the paper is organized as follows: Section 2 reviews related work; Section 3 defines the problem; Section 4 presents our proposed on-site planning algorithms; Section 5 describes the evaluation experiment; Sections 6 and 7 describe and discuss experimental results; Section 8 provides a summary and conclusions.

## 2. Related Work

### 2.1. Existing Work

Based on collected information, our proposed system recommends a sequence of places to visit according to the context of the various attractions, such as the user profile, congestion, and weather information. A number of prior studies [13–16] have investigated tourist recommendation methods based on the tourist destination characteristics and preferred route options.

In a study based on the static tourist context, Lim et al. [4] applied the PersTour algorithm [3] to find suitable PoIs. Using the average time spent in the candidate PoIs, together with indicators of popularity and the user preferences, the various spots are considered and a recommendation is made. Kurata and coworkers [17,18] developed a practical system called CT-Planner that analyzes user preferences and creates a tour route. However, such static-context recommendation systems are unable to respond to the dynamically changing conditions at the various destinations.

Several other studies have considered the dynamic tourist context [19–22]. Wu et al. [23] calculated the level of satisfaction of each spot by assigning probabilities to weather changes in P-Tour [12], which recommends a tour route likely to produce a high level of satisfaction. Jevinger and Persson [24] proposed an individual-optimized route method that takes into account the impact of public transport congestion. However, both of these recommendation systems handle only a single variable—weather or congestion—rather than a broader set of multiple dynamic factors. Moreover, they are not intended for on-site use.

Various other researchers have taken a multiobjective optimization approach and devised recommendation methods using machine learning [25,26]. Hirano et al. [27] proposed a system that recommends tour routes by taking into consideration the trade-off between the satisfaction obtained from touring and the resources (money, time, physical effort) consumed in travel and at the tourist site. Chen et al. [28] used PoI and route information as features in machine learning algorithms to recommend tour routes, modeling the tour recommendation problem as an orienteering problem [29,30], and proposed variations based on specific PoI visit sequences and PoI category constraints. However, these studies failed to consider that a visitor's satisfaction level changes with the time spent at each spot, and made recommendations based solely on the satisfaction level of the next spot.

There are several existing studies on tourism planning and tourist behavior in cities. McKercher et al. [31] showed differences in the behavior of first-time and repeat visitors during tourism. Caldeira et al. [32] also examined the effects of past experiences on tourists' temporal and spatial behavior. The results showed that first-time and repeat visitors differed in the scope of their activities and time spent in the areas they visited. Shoval et al. [33] also investigated the differences between users who were given incentives and those who were not. We show how the use of temporal and spatial data on tourist behavior can be used as a reference in formulating a tourism management plan for more rational management of tourist numbers.

As mentioned above, in the field of tourism, there have been various studies on the construction of tourism recommendation algorithms and on tourism planning and tourist behavior in cities. Since this paper focuses on the construction of a tourism recommendation algorithm, in the next

section, we present the problem of existing tourism recommendation algorithms and the position of our study.

*2.2. Problem of Existing Work and Positioning of Our Work*

In Section 2.1, we introduced the existing methods of recommending tourist spots and searching tourist routes. However, there are several problems with these. Table 1 shows a comparison between the existing studies and the Our work.

The first problem is that existing research assumes that tourists do not use the recommendation system onsite. Existing research considers that tourists are used before visiting a tourist site and does not consider or recalculate recommendations when visiting a tourist site. However, when a tourist actually visits a tourist site, depending on the weather, crowded conditions, and his/her mood at the time, they may want to consider the next spot to visit. Therefore, it is essential for tourists to use an on-site tourist spot recommendation system. There are several services (e.g., GoogleMaps [34]) that recommends nearby spots based on GPS information for on-site scenic spot recommendations. However, most of these services do not take into account the route situation. For example, if a tourist searches for a spot to visit, the recommended spot may be one that they have visited before. Systems such as P-tour [12,23] and CT-Planner [17,18] recommend effective routes by determining the start and finish points. However, these systems are intended to be used prior to a visit to a tourist site and are not designed for on-site use. If tourists stay longer than expected at a spot proposed by the system, the same route is calculated again, as the existing systems assume that the tourists use it before visiting the spot. Therefore, we need on-site tourist recommendation to recommend the next spot during the tour.

The second problem is that it does not take into account the dynamic context. Many existing methods take into account the user's preferences, which are static context. On the other hand, it does not take into account the weather information or the level of congestion. For example, depending on the weather conditions and other factors, whether one visits an indoor or an outdoor spot has a significant impact on the overall satisfaction with tourism. In an indoor spot such as a museum, satisfaction is unlikely to fluctuate regardless of weather conditions. However, at outdoor spots, such as shrines and temples, the level of satisfaction will be greatly affected by the weather conditions. Therefore, it is necessary to consider not only the static context (e.g., tourist preferences), but also the dynamic context (e.g., weather information and congestion).

The third problem is that it does not take into account the change in satisfaction with each spot depending on the time of the visit. For most existing methods, the satisfaction with each visited spot is constant regardless of the time of day visited. For example, if it is the season of autumn leaves and the spot is lit up with autumn leaves, the satisfaction level is higher if the tourist visits the spot when it is lit up than the usual satisfaction level [11]. In addition, in the case of route recommendation based on the model of the orienteering problem [35,36], they recommend a route that maximizes overall tourism satisfaction while spot satisfaction does not change with time of day. Since the level of satisfaction does not change regardless of the time of visit, tourists may have to visit the same spot again if they learn of a light-up during their visit. Therefore, it is necessary to take into account the change in satisfaction at the time of the visit at that spot.

The fourth problem is that it does not take into account the level of expected satisfaction. Many existing recommendation methods consider only the next tourist spot. For example, the next spot is recommended based on the level of satisfaction with the spot. If a highly satisfying spot is located far away from the current location, it may take a long time to get there, which may limit the number of spots to visit. This may lead to a decrease in overall satisfaction in the whole tour. In some cases, visiting three or more half-satisfied spots may lead to higher overall satisfaction than visiting a single satisfactory spot. Therefore, it is necessary to make a recommendation not only for the next spot, but also for the expected satisfaction of the next and subsequent spots.

**Table 1.** Compared to Existing work and Our work.

| Method | Next-POI | On-Site Next-POI Based Route | Reflect Preferences Static | Dynamic | Timeliness -Aware | Future Expection |
|---|---|---|---|---|---|---|
| Google Maps [34] | ✓ | | | | | |
| P-Tour [12,23] | | | ✓ | ✓(a) | | |
| CT-Planner [17,18] | | | ✓ | | | |
| Yuan et al [11] | | | ✓ | | ✓ | |
| The City Trip Planner [35,36] | | ✓ | ✓ | ✓(b) | | |
| This Work | ✓ | ✓ | ✓ | ✓(c) | ✓ | ✓ |

(a) only weather, (b) weather and congestion, (c) weather, congestion and additional point.

Most existing systems make recommendations based on the satisfaction of the next spot only, failing to give the best recommendation for the tour as a whole. Ideally, the best method would consider not only the tourist spot to be visited next, but also the expected satisfaction of the tourist spots that could be visited after that. Such a system must calculate the satisfaction for all tourist routes involving the next visited spot and the group of possible visited spots after that. However, this is an NP-hard problem, which means that deriving an optimal solution in a short time is not practical. This study shows that the knapsack problem, which is an NP-hard problem, is a special case of this problem. Given a limited amount of tourism time and given N types of spots, it is a problem to search for the next spot that maximizes the expected satisfaction of the tourist. The evaluation values for each of the N types of spots differ by the time spent and each time period, as well as the travel time between each spot. Here, the evaluation value of each spot at each time period is set constant, and the travel time between each spot is set to 0. In this problem, a number of spots are selected among N types of spots within the tourism time, and in order to maximize the sum of the evaluation values of the selected spots in the tourism time, the problem is a combination optimization problem of which spots to select. This problem is equivalent to the knapsack problem, which is NP-hard problem. In this study, we calculate the satisfaction level of the next spot and the expected satisfaction level of the next spot, instead of searching for the route with the highest overall satisfaction. This is why it is NP-hard problem. Therefore, the purpose of our study is to devise an algorithm that will find a good but not necessarily optimal solution relatively quickly. The system we proposed is a recommendation of on-site tourist spots on foot, taking into account the change in satisfaction with the temporal satisfaction of the spots and the expected satisfaction in the future. There are four main things we can achieve with our system: (i) on-site tourism recommendation. (ii) consideration of static and dynamic tourist contexts. (iii) consideration of changes in satisfaction with the time of the visit to each spot. (iv) consideration of possible future visits and expected satisfaction.

## 3. Preliminaries

In devising our approach, we define static and dynamic scores for the next spot and the future expected score after the next spot as variables to obtain on-site the overall tour satisfaction associated with visiting the next spot and all subsequent spots. A tour score $Tour(s, S, t)$ is given by the following equation, where the time of arrival at the set of possible spots $S$, the next spot $s$ and the arrival time of $s$ is $t$:

$$Tour(s, S, t) = SV(s) + DV(s, t) + EV(s, S - \{s\}, t + time(s)) \tag{1}$$

Here, $SV(s)$ and $DV(s, t)$ are the static and dynamic scores for spot $s$, respectively. (These are described in detail below.). $EV(s, S', t')$ is the maximum tour score obtained from touring spots in set

$S'$ from time $t'$ after touring spot $s$ (which we define later). The term $time(s)$ is the time spent at spot $s$ and the travel time from spot $s$ to $s'$.

The tour score calculated from this equation depends on the spot the tourist chooses as his/her next visit. The intent of this study is to calculate and present a tour score for each of the multiple tourist spot alternatives, allowing the tourist to choose the next spot to visit.

### 3.1. Static Score Component

We define the degree of matching between the user's preferences and the visiting spot s as the static score $SV(s)$. For these static scores, the evaluation method proposed by Lim et al. [4] is used.

### 3.2. Dynamic Score Component

The dynamic score $DV(s,t)$ at time $t$ for spot $s$ is calculated by the following equation:

$$DV(s,t) = TV(s,t) + CE(s,t) + WE(s,t) \qquad (2)$$

Here, $TV(s,t)$ represents an additional feature of spot $s$ at time $t$. For example, for a spot with a beautiful sunset or an exceptional night view, $TV(s,t)$ would take a positive value when time $t$ is in the evening or at night. $CE(s,t)$ is a term representing the level of congestion, taking a large value when spot $s$ is uncongested at time $t$ and a small value when it is congested. $WE(s,t)$ is a weather-related term that can be either positive or negative depending on the type of spot (e.g., indoor or outdoor) and the weather. These values are determined for each user's preference, spot, and situation.

### 3.3. Future Expected Score Component

The future expected score $EV(s,S',t')$ after visiting spot $s$ is defined recursively by the following equation:

$$EV(s,S',t') = \begin{cases} 0 & (\text{if } t' \geq T_{end}) \\ EV_{max} & (otherwise) \end{cases} \qquad (3)$$

$$EV_{max} = \max_{s' \in S' \wedge movet(s,s') + stayt(s') \leq T_{end}} \Big( SV(s') + DV(s',t') \\ + EV\big(s', S' - \{s'\}, t' + movet(s,s') + stayt(s')\big) \Big) \qquad (4)$$

Here, spot $s'$ is the next spot to visit after spot $s$. $T_{end}$ is the time of the tour's end, $movet(s,s')$ is the travel time from spot $s$ to $s'$, and $stayt(s')$ is the time spent (or stay time) at spot $s'$.

The future expected score $EV(s,S',t')$ is defined as the highest score that can be obtained for the group of spots that can be visited after visiting spot $s$ before the end of the tour $T_{end}$.

## 4. On-Site Tour Planning Algorithm

A description of the assumed environment for each of the three proposed algorithms is shown in Table 2. The spot list $Z$ shown in the table is a list of the tourist spots that will be visited based on the mobile application of the algorithm. Values for $\{Time, SpotID, Satisfaction\}$ for each tourist spot are stored in $Z$. The total satisfaction for the list of spots stored in $Z$ is the tour score. The top three tour scores are determined and presented to the user.

**Table 2.** Assumed environment used in the algorithm.

| Definition | Description |
|---|---|
| All spots set $S_{all}$ | $A, B, C, D, E, F, G, H, I$ |
| Set of visited spots $S_{visited}$ | $B, H$ |
| Set of unvisited spots $S$ | $A, C, D, E, F, G$ |
| Tourism time $T$ | 13:00–18:00 |
| Time slot width $tl$ | 1h |
| Current position $cp$ | $I$ |
| Current time $ct$ | 12:00 |
| List of spots to be visited $Z$ | $[ct, cp, 0]$ |
| List of spots to be visited on a temporary variable $Z_{tmp}$ | {} |

The static score $SV(s)$ is unchanged with time, while the dynamic score $DV(s,t)$ varies with time $t$ because of the variability of the measured values at each spot within each time period. To illustrate the application of the three proposed algorithms, each is applied in the assumed environment shown in Table 3.

**Table 3.** Score values of assumed environment used in the algorithm.

| | | Time | | | | | |
|---|---|---|---|---|---|---|---|
| | | **13:00** | **14:00** | **15:00** | **16:00** | **17:00** | **18:00** |
| | A | 7 | 3 | 4 | 5 | 6 | 7 |
| | B | 4 | 5 | 3 | 2 | 4 | 5 |
| | C | 4 | 5 | 6 | 7 | 9 | 6 |
| **Spot** | D | 4 | 5 | 4 | 3 | 2 | 6 |
| | E | 4 | 3 | 2 | 1 | 2 | 3 |
| | F | 7 | 7 | 6 | 4 | 3 | 2 |
| | G | 5 | 4 | 3 | 2 | 4 | 7 |
| | H | 4 | 5 | 4 | 3 | 2 | 1 |
| | I | 2 | 1 | 4 | 5 | 1 | 6 |

### 4.1. Setting Time Slot Width to Simplify the Problem

In the case of actual tourism, it may be desirable to produce a satisfaction value connected to the time spent at each spot using, for example, 1 min intervals. However, recommending a tour route in such a way would generate a huge amount of computation involved in assessing all the possible combinations of spots and visiting times. In addition to this computational challenge, our proposed algorithm is already expected to take more computational time than the usual tourism scheduling methods since it calculates satisfaction levels that can be expected in the future.

Therefore, rather than producing a value for each tourist spot each minute, we determine an evaluation value for each spot over a larger time interval, which allows us to calculate a sub-optimal solution relatively quickly under the assumption that only one spot is visited within a given interval. The proposed method calculates the total satisfaction value of a tourist spot for the specified time range as the sum of the static and dynamic scores.

The static score $SV(s)$ is unchanged with time, while the dynamic score $DV(s,t)$ varies with time $t$ because of the variability of the measured values at each spot within each time period. To illustrate

the application of the three proposed algorithms, each is applied in the assumed environment shown in Table 3.

*4.2. Overview of Three Algorithms*

As described below, the proposed algorithms for calculating tour scores are based on the greedy method.

**Time Series Greedy Algorithm (Algorithm A)** This is a greedy algorithm that considers only the evaluation values obtained for the next spot. An outline of the algorithm's application to the assumed environment is shown in Figure 1. In this algorithm, the arrival time at each spot in the set of unvisited spots $S$ is calculated based on the current location, taking into account the duration of the stay (stay time) and the travel time. Algorithm A identifies, in order, the spot with the maximum score considering only the next spot, and determines the top three tour scores (assuming $k = 3$).

**Whole Single Greedy Algorithm (Algorithm B)** This is a greedy algorithm that takes into account the evaluations obtained for all of the time slots. An outline of the algorithm's application to the assumed environment is shown in Figure 2. With Algorithm B, the top three tour routes are selected by considering the travel time to each spot and the duration of the stay (stay time) in the list of spots to be visited, $Z$.

**Whole Greedy Algorithm with Search Width (Algorithm C)** This is a greedy algorithm that considers the evaluations which obtained up to the top $k$ rank in all time slots. An outline of the algorithm's application to the assumed environment is shown in Figure 3. With Algorithm C, the top three tour routes are determined by recursively selecting spots within the top $k$ of the total tour time, taking into account the travel time and duration of stay for each spot in the list of spots to be visited, $Z$.

Details of each algorithm in pseudo-code are provided next, followed by examples of their application in the assumed environment.

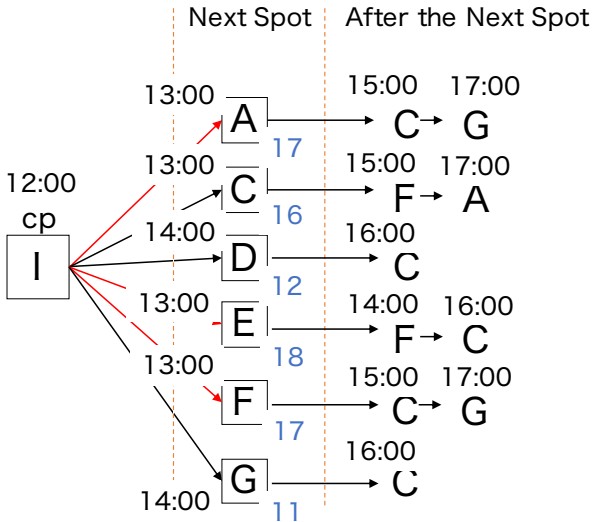

※ The figures in blue are the tour scores of the next spot visited.

**Figure 1.** Algorithm A.

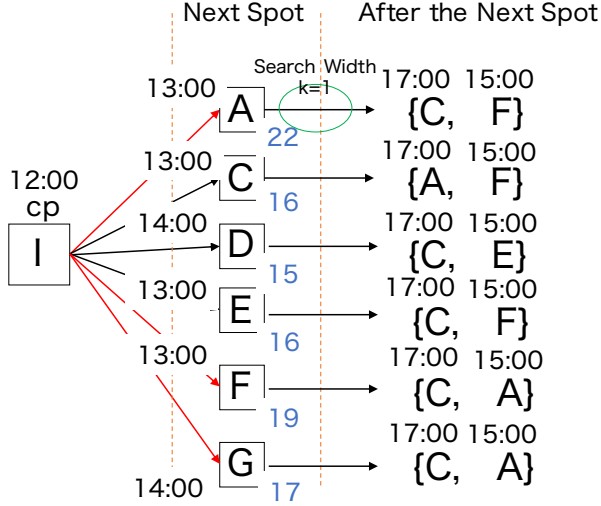

※ The figures in blue are the tour scores
of the next spot visited.

**Figure 2.** Algorithm B.

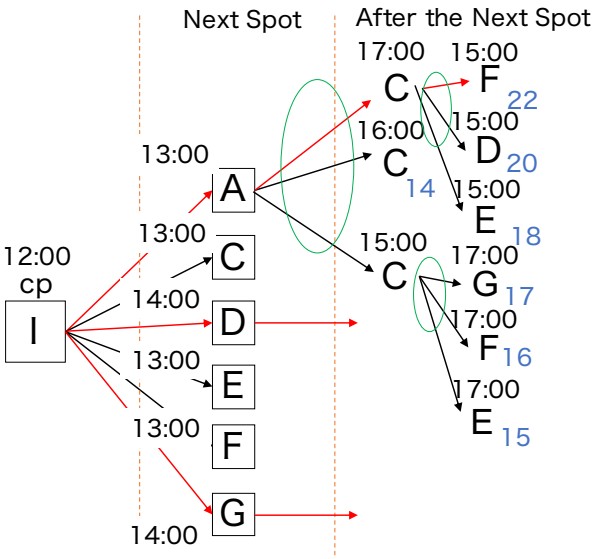

※ The figures in blue are the tour scores
of the next spot visited.

**Figure 3.** Algorithm C.

*4.3. Algorithm A (Time Series Greedy Algorithm)*

4.3.1. Details of Algorithm A

In Algorithm 1—P1 (**Main**), the algorithm outputs the results of the route with the top three tour scores (list of spots to be visited), which is the sum of the evaluation value of the next spot calculated in Algorithm 1—P2 (**GetOptRoutes**) and the future expected score calculated in Algorithm 1—P3 (**GetEVRoutes**), and the tour score of each route. In Algorithm 1—P2 (**GetOptRoutes**), the algorithm calculates the evaluation value (the sum of the static and dynamic scores) for each spot in the set of unvisited spots $S$. At this time, the selected tourist spot is stored in $Z$, the list of spots to be visited. Then, using the updated set of unvisited spots $S_{remain}$, the list of spots to be visited $Z$, and the tourism time $T$ as arguments, the algorithm calculates the tour score in Algorithm 1—P3 (**GetEVRoutes**). In Algorithm 1—P3 (**GetEVRoutes**), the algorithm selects the spot with the lowest tourism time

among the spots stored in $Z$. For each spot in set $S_{remain}$ (spots not yet visited), the evaluation value of the arrival time is calculated, taking into account the travel time and stay time for each spot. It then calculates the evaluation value of each spot and adds the spot with the highest evaluation value to $Z$, the list of spots to be visited. The spots added to $Z$ are removed from the set of unvisited spots $S_{remain}$. The same process is repeated until $T_{end}$ at the end of the tour, or until the set of unvisited spots is empty. The total evaluation value of the spots in $Z$ at the end of the tour is the tour score.

---

**Algorithm 1** Algorithm A (Time Series Greedy Algorithm)

---

**P1:** `Main()`
    **input**         **:** $S_{all}, S_{visited}, T, sp, k, cp, ct$
1    $S = S_{all} \setminus S_{visited}$
2    *Output GetOptRoutes* $(k, cp, ct, T, S, Z)$ as *Recommended_Route*

**P2:** `GetOptRoutes()`
    **input**         **:** $k, cp, ct, T, S, Z$
    **output**     **:** *Recommend_Route*
1    **foreach** *element s in S* **do**
2       Add *s* to $Z$ and remove *s* from $S_{remain}$
3       Calculate the tour score from *s* by `GetEVRoutes` $(T, S_{remain}, Z)$
4    **return** the $k$ largest routes from the next spot

**P3:** `GetEVRoutes()`
    **input**         **:** $T, S_{remain}, Z$
    **output**     **:** $R_{out}$
1    Select the lowest tourism time spot among the spots stored in $Z$
2    **foreach** *element ns in $S_{remain}$* **do**
3       Determine if it is possible to move to *ns* considering the time spent and travel time
4    Store the largest *ns* among the movable *ns* in $Z$
5    Recursively apply `GetEVRoutes` $(T, S_{remain}, Z_{tmp})$ to find the maximum tour score route
6    **return** the top route for tour score

---

### 4.3.2. Example of Algorithm A

To illustrate Algorithm A, we applied it to the assumed environment described in Table 3. At the start, the user's location is {12:00, *I*}. Algorithm A first calculates the evaluation value considering the travel time to each spot in the unvisited spot set $S_{remain}$. For example, if the tourist next visits spot *A*, i.e., {13:00, *A*}, the satisfaction score is 7. This is stored as {13:00, *A*, 7} in $Z$, the list of spots to be visited. Next, considering the time spent in {13:00, *A*} and the time spent traveling to each possible next spot, the maximum value of 6 occurs for spot *C* at 15:00. Thus, {15:00, *C*, 6} is stored in $Z$. Finally, considering the time spent in *C* and the travel time to each possible next spot, the maximum value of 4 occurs for spot *G* at 17:00, which means {17:00, *G*, 4} will be stored in $Z$. The tour ends with the time spent at {17:00, *G*}. The total satisfaction level stored in $Z$ is thus 17. This is the tour score. Ultimately, the tour score for each spot in each unvisited spot set $S_{remain}$ is calculated, and the top three tour scores are presented to the user.

### 4.4. Algorithm B (Whole Single Greedy Algorithm) and Algorithm C (Whole Greedy Algorithm with Search Width)

#### 4.4.1. Details of Algorithms B and C

Since Algorithms B and C are the same except for the search width $k$ ($k = 1$ and $k > 1$, respectively), the pseudo-code shown in Algorithm 2 is used to explain both Algorithms B and C. Algorithm 2—P1 (**Main**) is identical to Algorithm 1—P1 (**Main**). (Please refer to Section 4.3.)

In Algorithm 2—P2 (**GetOptRoutes**), the algorithm calculates the evaluation value (the sum of the static and dynamic scores) for each spot in the set of unvisited spots $S$. At this time, the selected spot is stored in $Z$, the list of spots to be visited. Next, it calculates the tour score in Algorithm 2—P3

(**GetEVRoutes**) as described below, with the search width $k$, current location $cp$, set of unvisited spots $S$, list of spots to be visited $Z$, and tourism time $T$ as arguments. In Algorithm 2—P3 (**GetEVRoutes**), the top $k$ evaluated values of arrival time, spot are selected using the sort result $TS$ calculated. We then employ GetEVRoutes recursively with the search width $k$, current location $cp$, set of unvisited spots $S_{remain}$, replicated list of temporary spots to be visited $Z_{tmp}$, and tourism time $T$ as arguments. $TS$ are sorted in descending order based on the tour time $T$, set of unvisited spots $S$, and list of spots to be visited $Z$. If more than one recurrence result is returned, the route with the best evaluation value among them is stored in $Z_{out}$. If there is no recurrence result, the result before the recurrence is stored in $Z_{out}$. When $Z_{out}$ is stored in $R_{out}$, if $R_{out}$ contains $k$ routes, the iteration is terminated and $R_{out}$ at that time is returned. The total evaluation value of the spots stored in the list of spots to be temporarily visited at the end of the tour is the tour score.

The search width used in the pseudo-code described here was set to $k = 1$ for Algorithm B and $k = 3$ for Algorithm C. (As noted earlier, $k > 1$ for Algorithm C.)

---

**Algorithm 2** Algorithm B (Whole Single Greedy Algorithm) and Algorithm C (Whole Greedy Algorithm with Search Width)

---

**P1:** `Main()`

    **input**           : $S_{all}, S_{visited}, T, sp, k, cp, ct$

1    $S = S_{all} \setminus S_{visited}$

2    *Output GetOptRoutes* $(k, cp, ct, T, S, Z)$ as *Recommended_Route*

**P2:** `GetOptRoutes()`

    **input**           : $k, cp, ct, T, S, Z$

    **output**        : *Recommend_Route*

1    **foreach** *element s in S* **do**

2       Add $s$ to $Z$ and remove $s$ from $S_{remain}$

3       Calculate the tour score from $s$ by `GetEVRoutes` $(k, cp, T, S, Z_{tmp})$

4    return the $k$ largest routes from the next spot

**P3:** `GetEVRoutes()`

    **input**           : $k, cp, T, S, Z$

    **output**        : $R_{out}$

    **temporal_variable**: $Z_{tmp}$

1    Create descending sorted sequences $TS$ using $T$ and $S$,considering $Z$

2    **foreach** *element ts in TS* **do**

3       Determine if it is possible to move to $ts$ considering the time spent and travel time

4       Add $ts$ to $Z$ and assign it to $Z_{tmp}$

5       Recursively `GetEVRoutes` $(k, cp, T, S, Z_{tmp})$ to find the maximum tour score route

6       **if** *there are k or more root results* **then**

7          `break`

8    return the $k$ routes for tour score

---

### 4.4.2. Example of Algorithms B and C

We applied Algorithm B to the environment described in Table 3. The user's start location is again {12:00, *I*}. Evaluation values are calculated taking into account the travel time to each spot in the set of unvisited spots $S$. For example, if the next spot to be visited is {13:00, *A*}, the satisfaction level will be 7. Consequently, {13:00, *A*, 7} is entered in $Z$, the list of spots to be visited. Next, among the remaining candidates, the highest satisfaction value, 9, is for spot *C* at 17:00. Thus, {13:00, *A*} to {17:00, *C*} are stored in the list of spots to be visited, since this pairing is feasible considering the travel time and stay time.

Now the highest value available is 7. However, given the {arrival time, spot} entries already stored in the list of spots to be visited, no spot with an evaluation value of 7 is feasible. The next highest evaluation value is 6, for spot *F* at 15:00 (i.e., {15:00, *F*}). Considering the travel time and stay time, it is feasible to travel from {13:00, *A*} to {15:00, *F*}, and from {15:00, *F*} to {17:00, *C*}. Therefore,

these entries are stored in $Z$. Since there are no more choices that can be added to the list, the tour is over. The total satisfaction level for the items stored in $Z$ is 22. This is the tour score for a tour starting with $A$ as the next spot visited. The tour score for each spot in the set of unvisited spots $S$ is similarly calculated, and the top three tour scores are presented to the user.

Algorithm C can be applied to the same hypothetical environment. As before, the user's current location is {12:00, $I$}, and the evaluation value is calculated taking into account the travel time to each spot in the set of unvisited spots $S$. For example, if spot $A$, with a satisfaction value of 7, is the first spot to be visited, then {13:00, $A$, 7} is entered in $Z$, the list of spots to be visited. After spot $A$, the top three evaluated values in terms of the total tour time excluding the time spent in spot $A$ are {17:00, $C$, 9}, {16:00, $C$, 7}, and {15:00, $C$, 6}. Given the travel time and stay time, it is feasible to proceed from {13:00, $A$} to {17:00, $C$}. Consequently, these entries are stored in the temporary visitation list $Z_{tmp}$.

The top three evaluation values after taking into account the travel time from spot $A$ to spot $C$ and the time spent at spot $C$ are {15:00, $F$, 6}, {15:00, $D$, 4}, {15:00, $E$, 2}. For all of these, it is feasible to visit {13:00, $A$} and also to visit {17:00, $C$}. Storing the list of spots to be visited takes place at the end of the tour time for any route. When the tour score is calculated based on each evaluation value, the maximum route is added to $R_{out}$. In this case, the maximum route is produced by adding {15:00, $F$, 6} to the temporary visitation list, $Z_{tmp}$. The same can be done with {16:00, $C$, 7} or {15:00, $C$, 6}. The largest route in $R_{out}$ is added to the tour score route when there are three routes in $R_{out}$. At this point, the tour score for {13:00, $A$} can be calculated. Ultimately, the same procedure is applied for each spot in the set of unvisited spots $S$, calculating the tour score for each spot and presenting the top three tour scores to the user.

## 5. Experiment

### 5.1. Objective of the Experiment

To test the effectiveness of the proposed algorithms, they were applied to an area in Higashiyama, Kyoto, Japan, containing 20 PoIs (Table 4, Figure 4), and tour scores were produced and analyzed. In addition, since this study assumes an on-site navigation capability, the computation times required to produce the tour scores were recorded.

**Table 4.** List of Symbols.

| Symbol | Description | Symbol | Description |
|--------|-------------|--------|-------------|
| IK | Ishibe-Koji | SGR | Shore-in Gate Ruins |
| RNT | Rokuhara Mitsuji Temple | KM | Kyoto Minamiza |
| KCM | Kyoto Culture Museum | KYT | Kiyomizu Temple |
| CIT | Chion-in Temple | CHT | Chorakuji Temple |
| YK | Yasui Konpiragu | MP | Maruyama Park |
| NM | Nishiki Market | KNT | Kenninji Temple |
| KRGS | Kyoto Ryozan Gokoku Shrine | YS | Yasaka Shirine |
| RD | Rokkakudo | TT | Tohukuji Temple |
| HS | Hanamikoji Street | NZ | Ninenzaka |
| KDT | Kodaiji Temple | SSD | Sanju Sangen Do |

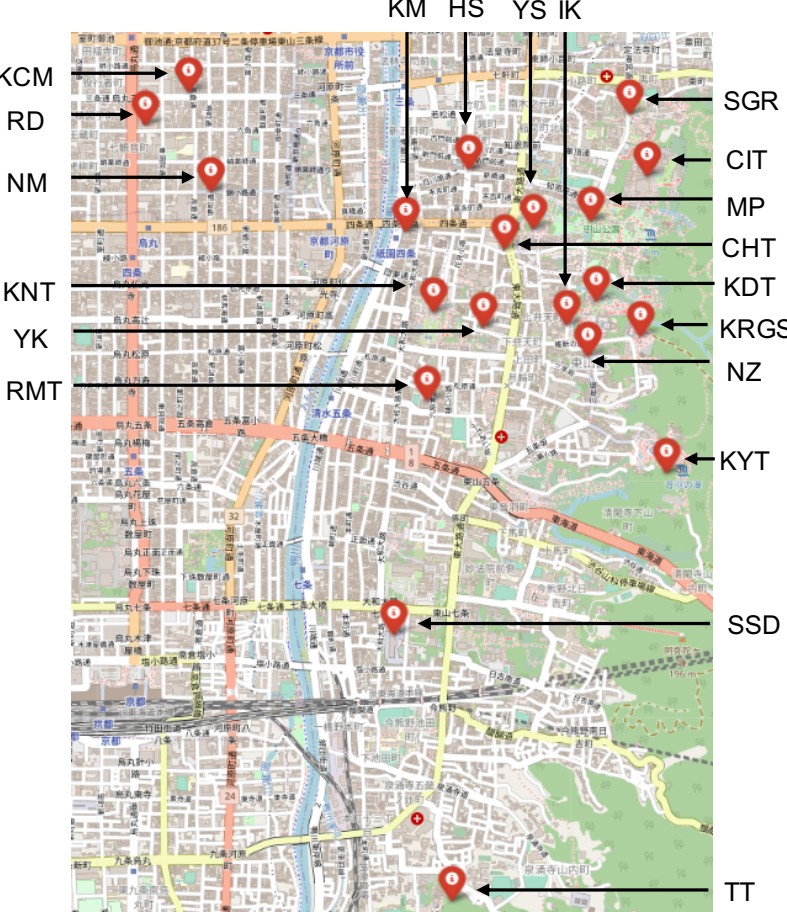

**Figure 4.** 20 PoIs in Higashiyama.

## 5.2. Contents of the Experiment

The proposed algorithms were written in Python and executed on a machine with an Intel Core i5 2.3 GHz CPU with 8.0 GB memory and amacOS Catalina OS. The travel time between each spot and the stay time at each spot were taken from tourist information magazines [37,38] and from the Google Maps API (for values not included in the tourist information magazines). The static score was obtained from Google Maps. A static score $SV$ was assigned to each spot, scaled from 1 to 5. The dynamic score $DV$ consists of additional points $TV$, congestion level $CE$, and weather-related term $WE$. $TV$, $CE$ and $WE$ were calculated as follows. $TV$ was calculated based on event information (lighting at night) based on the Kyoto tourist information magazines. $TV$ were assigned for each spot at each time, scaled from 0 to 2. For three spots (KDT, KYT, CIT), the time period 17:30–21:00 was assigned a value of +2, while the value for $TV$ (the special feature variable) was 0 to 2. Four spots (RD, MP, SGR, TT) were assigned a value of +1 because they were featured in a tourist information magazine with text only without a photograph of illuminated maple leaves. For the other spots, the assigned value was 0. For CE, we used congestion data for each spot from Google Maps on a Sunday, as of 2 April 2020. A visualization of the congestion levels is shown in Figure 5. The congestion values are for every 30 min. To produce a value for every 10 min, the 30 min values were replicated and a scale transformation to produce values was applied to the inverse of the replicated values. CE were assigned for each spot at each time, scaled from 0 to 2. $WE$ were assigned for each spot at each time, scaled from –1 to 1. (For an outdoor spot, the assigned value is -1 if it rains and +1 if it is sunny. For indoor spots, the assigned value is +1 if it rains and 0 if it is sunny.). The heatmap for the total score calculated from the static and dynamic scores for each time period for each spot is shown in Figure 6.

On the other hand, the way to estimate the dynamic score $DV(s,t)$ is also important in order to make our proposed method practical We think several methods for estimating the dynamic score $DV$ in real time. The additional point $TV$ is calculated using participatory sensing [39]. By using participatory sensing, tourists can share the information of points (e.g., information on lighting at night) at actual tourist spots. Congestion point $CE$ is calculated based on the congestion level of each tourist spot based on the yahoo congestion radar [40]. The weather information WE is calculated based on the values for each spot type (indoor or outdoor) based on the weather information obtained from the OpenWeatherAPI [41]. These data are collected and updated at any time. The system re-searches the route when the tourist situation (e.g., unexpected events) changes significantly, or when the user makes another planning request (When user is deciding on user's next spot.). The implementation of those function is future work.

The tour scores produced by the three proposed algorithms were compared with the total value of model routes described in the tourist information magazines. Because the system is assumed to allow on-site navigation, a target computation time of under a few minutes was considered realistic [42].

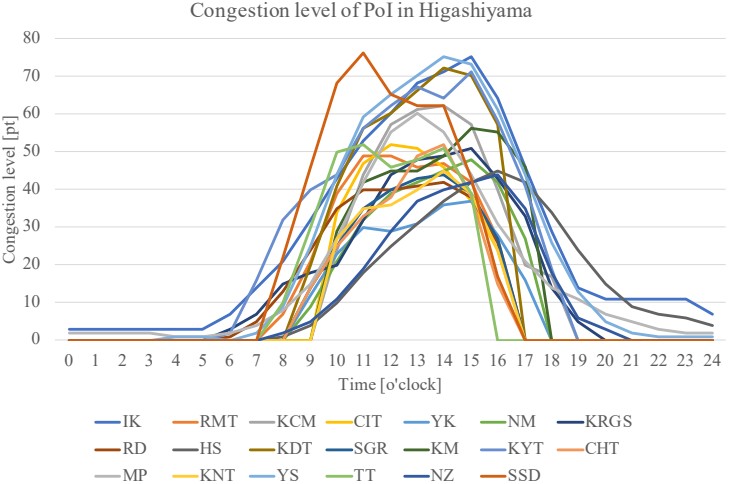

**Figure 5.** Congestion level of 20 PoIs in Higashiyama.

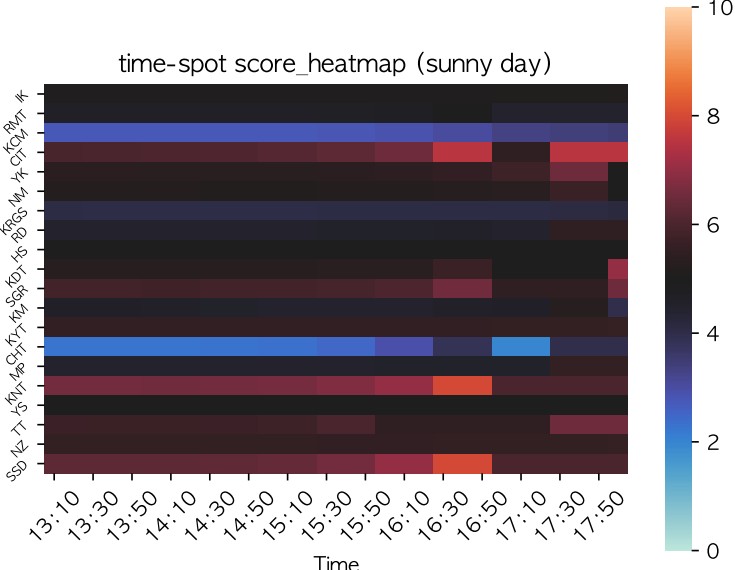

**Figure 6.** Score for each spot and time slot (sunny day).

## 6. Results

The experimental environment was assumed to be a sunny autumn day. The tour time was assumed to be 5 h, from 13:00 to 18:00. The time slot width was 10 min. The departure point was Gion Station, and there were no previously visited spots. The three proposed algorithms were run five times in the experimental environment. The output solutions and computation times are described below.

### 6.1. Output Solutions

Table 5 shows the output solutions for the top three tourist routes when each algorithm was applied to the experimental environment. Based on a comparison of tour score values, Algorithm C performed the best and Algorithm B performed the worst.

A comparison of Algorithms A and B reveals several interesting results. With Algorithm A, visits are not made to spots with the highest evaluation value (16:30, SSD) and (16:30, KNT) in Figure 3. Here, popular spots are not visited at the time of their highest evaluation value since spots with high evaluation values are randomly visited, taking into account the stay and travel times. Because of this, it is more likely that spots are visited at a better time with Algorithm B. However, in terms of overall satisfaction, Algorithm A was superior.

Comparing Algorithms A and C, it can be seen that there is not much difference in the spots to be visited, but the time of day for the visits is substantially different. Algorithm C, which considers the top $k$ spots among the overall evaluation values, was able to schedule visits at a better time, and thus the expected satisfaction was superior.

Comparing Algorithms B and C, Algorithm C considers the top $k$ spots having a large overall evaluation value and also considers the spots that can be visited in the future, making Algorithm C's tour scores superior. However, Algorithm B resulted in superior score values when KNT is visited.

From the above, the three proposed algorithms work effectively in achieving their intended purpose. While Algorithm C produces the highest overall tour scores, Algorithm B tends to produce visits to high-scoring spots at their best time of day.

### 6.2. Computation Times

The computation times for each algorithm for each of the five runs are given in Table 6. The results show average computation times of $1.9 \pm 0.1$ (s), $2.0 \pm 0.1$ (s), $27.0 \pm 1.8$ (s) for Algorithms A–C, respectively. Algorithms A–C were on-site practical in terms of computation time, and had satisfactory performance. All of the proposed algorithms were able to output a solution within one minute, which means that they can be used on-site.

**Table 5.** Results for the three proposed algorithms.

| | Route | Result | Tour_Score | Count_Spot | Mean |
|---|---|---|---|---|---|
| **Algorithm A** | Best | [13:20, KDT, 5.3], [14:00, KNT, 6.5], [14:50, SSD, 6.3], [16:10, CIT, 6.5], [16:50, SGR, 6.5], [17:30, MP, 5.6], [17:50, IK, 5.1] | 41.9 | 7 | |
| | Second | [13:10, KNT, 6.6], [14:00, SSD, 6.3], [15:20, CIT, 6.1], [16:00, SGR, 6.0], [16:50, KDT, 5.7], [17:30, MP, 5.6], [17:50, IK, 5.1] | 41.4 | 7 | 41.6 |
| | Third | [13:20, SSD, 6.3], [14:30, KNT, 6.6], [15:20, CIT, 6.1], [16:00, SGR, 6.0], [16:50, KDT, 5.7], [17:30, MP, 5.6], [17:50, IK, 5.1] | 41.4 | 7 | |
| **Algorithm B** | Best | [13:10, HS, 5.0], [13:40, SGR, 5.8], [14:30, IK, 5.1], [15:00, SSD, 6.4], [16:30, KNT, 8.0], [17:30, CIT, 8.0] | 38.3 | 6 | |
| | Second | [13:10, IK, 5.1], [13:40, SGR, 5.8], [14:30, HS, 5.1], [15:00, SSD, 6.4], [16:30, KNT, 8.0], [17:30, CIT, 8.0] | 38.3 | 6 | 38.3 |
| | Third | [13:10, YS, 5.0], [13:40, SGR, 5.8], [14:30, IK, 5.1], [15:00, SSD, 6.4], [16:30, KNT, 8.0], [17:30, CIT, 8.0] | 38.3 | 6 | |
| **Algorithm C** | Best | [13:10, YS, 5.0], [13:40, SGR, 5.8], [14:30, CHT, 2.3], [15:00, KNT, 6.6], [15:50, KDT, 5.3], [16:30, CIT, 7.5], [17:10, HS, 5.0], [17:30, MP, 5.6], [17:50, IK, 5.6] | 48.3 | 9 | |
| | Second | [13:20, SGR, 5.8], [14:00, CHT, 2.3], [14:30, KNT, 6.6], [15:20, KDT, 5.3], [16:00, YS, 5.0], [16:30, CIT, 7.5], [17:10, HS, 5.0], [17:30, MP, 5.6], [17:50, IK, 5.6] | 48.2 | 9 | 47.8 |
| | Third | [13:10, KNT, 6.6], [14:00, SSD, 6.3], [15:20, SGR, 5.9], [16:00, YS, 5.0], [16:30, CIT, 7.5], [17:10, HS, 5.0], [17:30, MP, 5.6], [17:50, IK, 5.6] | 47.0 | 8 | |

**Table 6.** Computation times for the three proposed algorithms.

| | Computation Time (s) | | | | | |
|---|---|---|---|---|---|---|
| | **First** | **Second** | **Third** | **Fourth** | **Fifth** | **Mean** |
| **Algorithm A** | 1.8 | 2.0 | 1.9 | 1.8 | 1.8 | $1.9 \pm 0.1$ |
| **Algorithm B** | 2.2 | 1.9 | 1.8 | 2.0 | 2.0 | $2.0 \pm 0.1$ |
| **Algorithm C** | 29.9 | 25.5 | 26.3 | 25.1 | 28.0 | $27.0 \pm 1.8$ |

*6.3. Setting the Width in Algorithm C*

In the example above, the search width in Algorithm C was set to $k = 3$. To test the effect of the search width, a variable search width was considered. Figure 7 shows the computation time and tour score (overall tour satisfaction) when the search width $k$ is assigned values between 1 and 5. As can be seen, the computation time increases exponentially with increasing $k$, while the tour score increases roughly linearly and then saturates. The reason for this is that, as the search width increases, more combinations of spots and time periods that can be visited in the future are searched.

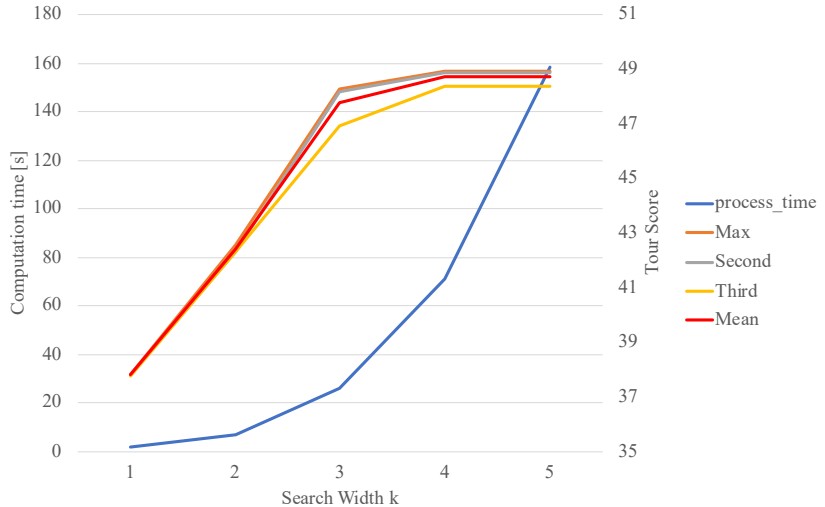

**Figure 7.** Computation time and tour score associated with search width *k*.

## 6.4. Comparison with Model Routes

To assess the comparative performance, the output solutions from the proposed algorithms were compared with routes suggested in the two tourist information magazines used as references in the study [37,38]. The results are shown in Table 7. The calculations are based on the evaluated values in the experimental environment using a tour time of approximately 5 h.

In terms of mean tour scores, Algorithms A–C (with scores of 41.6, 38.3, and 47.8, respectively) all outperform Model Route 1 (27.3) In Model Route 1, the tour score is low, primarily because tourists tend to select the most famous spots covered in the tourist information magazines (meaning a long stay) and do not necessarily visit spots at the times when their evaluation values are highest.

**Table 7.** Scores for model routes.

| Route | Result | Tour_Score | Count_Spot |
|-------|--------|------------|------------|
| **Model Route 1** | [13:20, KYT, 5.5], [14:40, KDT, 5.3], [15:20, MP, 4.6], [16:20, CIT, 6.5], [17:30, SGR, 5.5] | 27.3 | 5 |
| **Model Route 2** | [13:10, YS, 5.0], [13:30, IK, 5.1], [13:50, KRGS, 3.1], [15:20, KDT, 5.3], [16:00, CHT, 3.4], [16:30, MP, 4.9], [16:50, CIT, 7.0], [17:30, SGR, 5.5] | 39.8 | 8 |

On the other hand, while Model Route 2 (39.8) produced an inferior score to Algorithm A (41.6) or Algorithm C (47.8), it produced a higher score than Algorithm B (38.3). A closer look, however, revealed that while Model Route 2 was superior to the route produced by Algorithm B in terms of overall tour score, Algorithm B produced a score of 8 for visiting CIT at 17:30, while Model Route 2 produced a score of 7.5 for visiting the same site at 16:50; that is, Algorithm B found a better time to visit CIT. The same is true for SGR. The implication is that using the overall tour score or overall satisfaction as a clear indicator of superiority or inferiority may not be appropriate in all cases.

## 6.5. Output Solution When the Experimental Environment Is Changed

In the previous sections, we have presented the output solution of the proposed algorithms on sunny weather. In this section, we show the output solution of the proposed algorithm in rainy weather, with the time period one month in advance. For the sake of the space limitation, we show the output solution only for the proposed Algorithm C. The schedule was shifted one month in advance, and the tour time is assumed to be from 13:00 to 18:00, and the tourism spots are applied to an area in Higashiyama, Kyoto, Japan, containing 20 PoIs (Table 3, Figure 2). In this section, the value of the additional point TV set in Section 5.2 is different from the value set in the previous section because

the time is moved forward by one month. Since the maple leaves are not illuminated and a special exhibition is held at KCM, we give KCM +2 only. For the other spots, the assigned value was 0. The weather information WE set in Section 5.2 is different from the previous sections (+1 for indoor spots and −1 for outdoor spots) in order to assume rainy weather.

The heatmap of the total score calculated from the static and dynamic scores for each spot at each time is shown in Figure 8. The results in Figure 8 show that the scores for the outdoor spots at each time of the day are smaller than the heat map (Figure 6) for the sunny day. On the other hand, the indoor spots (KM, KCM) showed a larger score.

Table 8 shows the results of applying the proposed Algorithm C to the rainy weather environment. The results in Table 8 show that the indoor spot, KCM, was selected in rainy weather. A similar indoor spot, KM, has a long average stay of 2.5 h, so its selection will reduce the number of spots that can be visited in the future, resulting in a smaller tour score. As a result, KM is less likely to be selected.

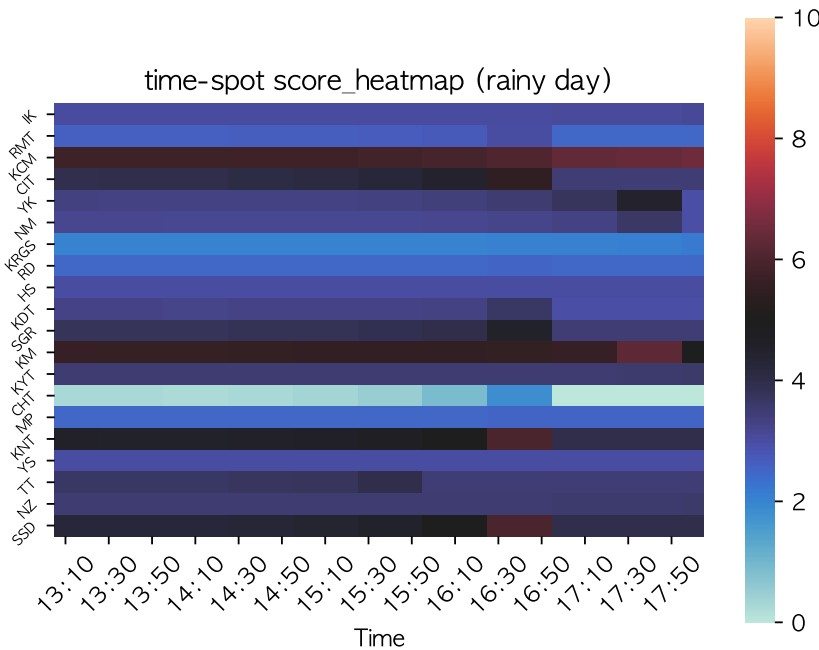

**Figure 8.** Score for each spot and time slot (rainy day).

**Table 8.** Results for the proposed algorithm C in rainy.

|  | Route | Result | Tour_Score | Count_Spot | Mean |
|---|---|---|---|---|---|
| **Algorithm C** | Best | [13:10, HS, 3.0], [13:30, YS, 3.0], [14:00, CIT, 4.0], [14:50, KCM, 5.8], [16:40, KNT, 6.0], [17:30, IK, 3.1], [17:50, MP, 2.6] | 27.5 | 7 | 27.5 |
|  | Second | [13:10, MP, 2.5], [13:30, YS, 3.0], [14:00, CIT, 4.0], [14:50, KCM, 5.8], [16:40, KNT, 6.0], [17:30, IK, 3.1], [17:50, HS, 3.0] | 27.5 | 7 |  |
|  | Third | [13:10, YS, 3.0], [13:40, HS, 3.0], [14:00, CIT, 4.0], [14:50, KCM, 5.8], [16:40, KNT, 6.0], [17:30, IK, 3.1], [17:50, MP, 2.6] | 27.5 | 7 |  |

## 7. Discussion

The reason that Algorithm A produces less satisfaction than Algorithm C is derived from the fact that the score values of the various spots differ depending on the time of the visit. In contrast to

Algorithm A, Algorithm C selects a spot from the top three evaluation values by considering spots that can be visited in the future, enabling it to find a better time to visit a spot.

One of the reasons that the tour scores of Algorithm B turned out to be inferior to those of the two other algorithms is the presence of substantially more free time in the Algorithm B result, as indicated by the red areas in Figure 9 (note that Figure 9 shows only the highest scoring route for each algorithm). As shown, the total free time in the Algorithm B result is 30 min, whereas there is no free time at all in the results of the other two algorithms. Unlike Algorithm B, where the search width is 1, Algorithm C uses a search width of 3, which reduces the problem of fragmentation encountered by Algorithm B.

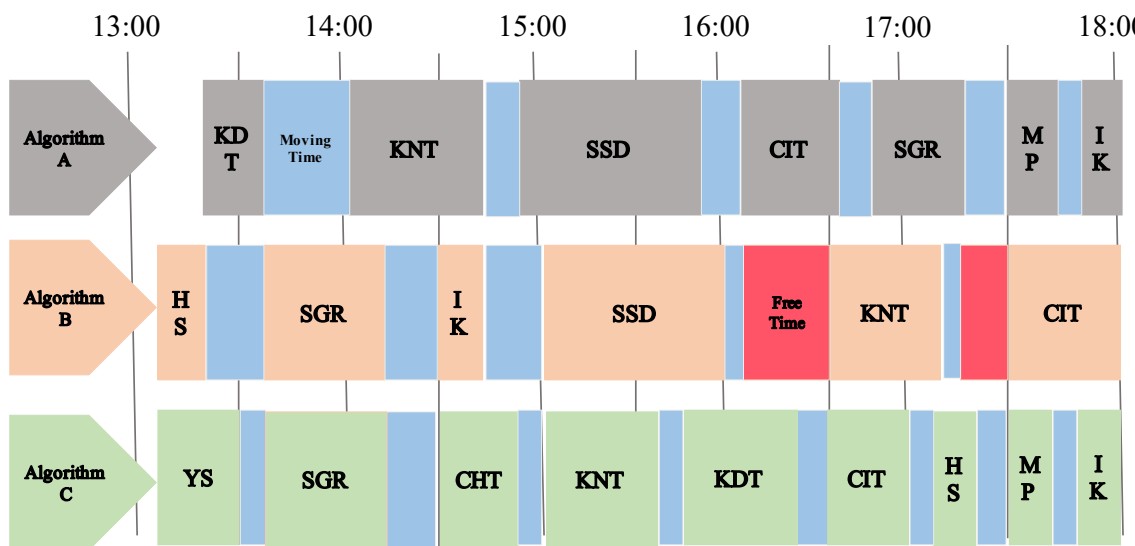

**Figure 9.** Free time for each proposed algorithm.

## 8. Conclusions and Future Work

In this paper, we defined a tour score consisting of three elements -static score, dynamic score, and future expected score- for the on-site tourist route search problem and proposed the Time Series Greedy Algorithm (Algorithm A), Whole Single Greedy Algorithm (Algorithm B), and Whole Greedy Algorithm with Search Width (Algorithm C) to solve the problem.

We applied the algorithms to 20 PoIs in Higashiyama, Kyoto, Japan, and evaluated the quality of the solution (tour score) and the computation time for each algorithm. As a result, we found that Algorithms A and B produced tour scores in realistic computation times, (b) Algorithm C produced the highest tour scores, and the computation time was still realistic, and (c) the computation time for Algorithm C increases exponentially as the width increases. The experimental results confirmed that the three proposed algorithms can output quasi-optimal solutions with trade-offs in computation time. The computation times when Algorithms A–C were applied to the same experimental environment were $1.9 \pm 0.1$, $2.0 \pm 0.1$, and $27.0 \pm 1.8$ s, respectively.

In the future, we plan to verify the effectiveness and efficiency of the proposed algorithms in multi-day travel [43] and multiple areas [36], because it is possible to assume not only a single area but also tourism in other area. In addition, since this paper focuses on the construction of the on-site tourism recommendation algorithms, we do not refer to a method for estimating the dynamic tourist context in real time. We plan to apply the real-time estimation of the dynamic tourist context to the proposed algorithms.

**Author Contributions:** Conceptualization, S.I. and K.Y.; methodology, S.I. and K.Y.; software, S.I. and Y.M.; validation, S.I. and Y.M.; formal analysis, S.I. and H.S.; investigation, S.I. and Y.M.; resources, S.I. and M.H.; data curation, S.I. and H.S.; writing—original draft preparation, S.I. and M.H.; writing—review and editing, Y.M., H.S. and K.Y.; visualization, S.I.; supervision, H.S. and K.Y.; project administration, S.I. and K.Y; funding acquisition, H.S. and K.Y. All authors have read and agreed to the published version of the manuscript.

**Funding:** This research received no external funding.

**Conflicts of Interest:** The authors declare no conflict of interest. The founding sponsors had no role in the design of the study; in the collection, analyses, or interpretation of the data; in the writing of the manuscript; or in the decision to publish the results.

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
