# Peer review of "Timeliness-Aware On-Site Planning Method for Tour Navigation"

_smartcities, doi:10.3390/smartcities3040066_

Round 1
Reviewer 1 Report
Thanks for the invitation.
The article offers a good overview about optimizing time for visits.
The methodological session is very complete and I think it is the most powerful part of the paper.
The bibliographic review session can improve because the authors go very directly to their central theme, forgetting to make the life citations. I recommend checking this part.
I found the paper easy to read, although it is very dense in terms of methodology.
It is necessary to quote when it is said:
-85 "A number of prior studies have investigated tourist recommendation"
-94 "Several other studies have considered the dynamic tourist context."
-100 "Various other researchers" (and more).
I recommend bringing the tables closer to the text.
Finally, although the bibliography is appropriate, I think it could be more up to date. In this journal there are studies on this topic.
Congratulations on the work, I think when it is completed it will be a great contribution to the academy.
Reviewer 2 Report
The proposed article is a good analysis of the use of tourism applications in urban tourism. The authors verified the proposed approach to using the application using three algorithms. A very detailed statistical analysis allowed to draw correct conclusions. Large tourist traffic also requires a modern approach and the use of new research methods. The proposed article is interesting and should be published.
Reviewer 3 Report
The topic developed in the paper is interesting as it proposes a model for solving some of the existing problems in contemporary cities. Nevertheless, I found the paper to technical and I would suggest to include a more practical approach.
In the revision of previous works, the authors focus the attention on similar works based on recommendation of tourist spots, but should be interesting to have a wider look. There are several studies focusing on the behaviour of tourists when visiting cities and how to assess them in order to improve management plans. For example, Shoval proposed a method of tracking tourists in heritage cities to better understand their behaviour. I think that it could enrich much the paper to include a section reviewing some of the theories on tourism planning in cities and tourists behaviour.
The same could be said in the results and conclusions. Even the experiment done is interesting, a more applied vision could be interesting. To what extend the model that the authors propose can help improving tourism management? Can be the model tested with visitors? How are they going to access the information, thought a mobile app?
